# Investigation of Clinical Features and Association between Vascular Endothelial Injury Markers and Cytomegalovirus Infection Associated with Thrombotic Microangiopathy in Patients with Anti-Neutrophil Cytoplasmic Antibody (ANCA)-Associated Vasculitis: Case-Based Research

**DOI:** 10.3390/ijms25020812

**Published:** 2024-01-09

**Authors:** Takayuki Nimura, Daiki Aomura, Makoto Harada, Akinori Yamaguchi, Kosuke Yamaka, Takero Nakajima, Naoki Tanaka, Takashi Ehara, Koji Hashimoto, Yuji Kamijo

**Affiliations:** 1Department of Nephrology, Shinshu University School of Medicine, 3-1-1, Asahi, Matsumoto 390-8621, Japan; 21hm122a@shinshu-u.ac.jp (T.N.); aomura91@gmail.com (D.A.); ymgc@shinshu-u.ac.jp (A.Y.); kyamaka423@gmail.com (K.Y.); khashi@shinshu-u.ac.jp (K.H.); 2Department of Metabolic Regulation, Shinshu University School of Medicine, 3-1-1, Asahi, Matsumoto 390-8621, Japan; nakat@shinshu-u.ac.jp; 3Center for Medical Education and Training, Shinshu University School of Medicine, 3-1-1, Asahi, Matsumoto 390-8621, Japan; 4Department of Global Medical Research Promotion, Shinshu University Graduate School of Medicine, 3-1-1, Asahi, Matsumoto 390-8621, Japan; naopi@shinshu-u.ac.jp; 5International Relations Office, Shinshu University School of Medicine, 3-1-1, Asahi, Matsumoto 390-8621, Japan; 6Research Center for Social Systems, Shinshu University, 3-1-1, Asahi, Matsumoto 390-8621, Japan; 7Department of Pathology, Shinshu University School of Medicine, 3-1-1, Asahi, Matsumoto 390-8621, Japan; eharat@shinshu-u.ac.jp

**Keywords:** anti-neutrophil cytoplasmic antibody, anti-neutrophil cytoplasmic antibody-associated vasculitis, cytomegalovirus, sulfatides, thrombotic microangiopathy

## Abstract

Anti-neutrophil cytoplasmic antibody-associated vasculitis (AAV) can occasionally trigger thrombotic microangiopathy (TMA). Cytomegalovirus (CMV) may be reactivated during intensive immunosuppressive therapy for AAV and cause TMA. Therefore, we aimed to evaluate the clinical features of and the association between vascular endothelial injury markers and TMA due to CMV in patients with AAV. A 61-year-old female was diagnosed with AAV and severe kidney injury. Immunosuppressive therapy gradually improved her symptoms and laboratory findings. However, 2 weeks after induction therapy initiation, she exhibited altered consciousness, a significant decrease in platelet count, and hemolytic anemia, resulting in a TMA diagnosis. Plasma exchange did not improve TMA findings and routine screening test revealed CMV infection. Ganciclovir injection improved the infection and TMA findings. Consequently, we diagnosed her with CMV-induced TMA. Both AAV and CMV may induce severe vascular endothelial injury, potentially leading to TMA development. CMV-induced TMA should be considered when TMA develops during induction therapy against AAV. Moreover, of the three serum markers of vascular injury—serum sulfatides, soluble thrombomodulin, and pentraxin 3—serum sulfatides may be associated with the development of TMA, and a high level of soluble thrombomodulin may be associated with the development of CMV viremia during the clinical course of AAV.

## 1. Introduction

Anti-neutrophil cytoplasmic antibody (ANCA)-associated vasculitis (AAV) is an autoimmune disease that causes severe inflammation in small vessels [1]. AAV can also result in multi-organ dysfunction, including glomerulonephritis and interstitial pneumonitis, often resulting in a poor prognosis [2,3]. Therefore, early initiation of intensive immunosuppressive therapy is recommended to improve the prognosis. Thrombotic microangiopathy (TMA) is characterized by the formation of platelet thrombi in capillaries and arterioles, induced by vascular endothelial injury [4], and typically exhibits a triad of hemolytic anemia, thrombocytopenia, and acute kidney injury. Moreover, 4–26% of patients with AAV have concurrent TMA [5,6] and experience the worst outcomes [7]. While vascular endothelial injury caused by AAV is considered to contribute to the onset of TMA, the precise cause for the frequent co-occurrence of TMA and AAV remains unclear.

Cytomegalovirus (CMV) is recognized as a cause of opportunistic infection in patients undergoing immunosuppressive therapy. Patients with AAV are also known to be at high risk of CMV reactivation and subsequent organ injury [8]. Additionally, CMV is known to infect vascular endothelial cells that may be associated with severe CMV infection, resulting in the development of TMA [9].

The relationship between several vascular endothelial injury markers and AAV has been investigated [10,11,12]. Soluble thrombomodulin and pentraxin 3 are well-known vascular endothelial injury markers. Additionally, we recently reported novel vascular injury markers, serum sulfatides, which are associated with the pathological activity of kidney lesions in patients with AAV [13]. Sulfatides are one of the glycosphingolipids measured using matrix-assisted laser desorption ionization–time-of-flight mass spectrometry (MALDI-TOF MS), and their low levels may indicate not only severe vascular endothelial injury but also platelet activation. Therefore, serum sulfatide levels have the potential to act as markers for TMA, similar to other vascular endothelial markers such as soluble thrombomodulin and pentraxin 3.

Herein, we report our case-based research conducted to identify the clinical features of and the association between vascular endothelial injury markers and CMV infection-associated TMA in patients with AAV. First, we aimed to investigate the details of the clinical course of CMV infection-associated TMA in a current patient with AAV. Second, we aimed to evaluate the three vascular endothelial markers—serum sulfatides, soluble thrombomodulin, and pentraxin 3—as markers for TMA by comparing their levels in donor candidates of living kidney transplantation as a control, patients with AAV, and the current case, respectively. Thirdly, because latent infection of CMV in vascular endothelial cells may be associated with the development of TMA, we aimed to statistically analyze the association between these three vascular endothelial markers and CMV viremia.

## 2. Results

### 2.1. Clinical Course of CMV Infection Associated with TMA in a Patient with AAV

A 61-year-old female presented to a general hospital complaining of systemic fatigue persisting for 3 weeks. She also experienced a mild fever, and a computed tomography (CT) scan revealed a right lung consolidation. She was admitted to the hospital on the suspicion of bacterial pneumonia, and treatment with antibiotics was initiated. However, her symptoms and clinical findings did not improve. Seven days after treatment initiation, she suddenly developed bilateral hearing impairment, and a laboratory test revealed an acute elevation of her serum creatinine (Cre) level (rising from 1.9 mg/dL at admission to 2.5 mg/dL), accompanying proteinuria and hematuria. Additionally, her serum myeloperoxidase (MPO)-ANCA tested positive, and she was transferred to our hospital for further assessment and treatment.

She had a history of hyperuricemia and was taking 50 mg/day of allopurinol. She had no allergies. Clinical examination revealed a blood pressure of 128/61 mmHg, a heart rate of 93 beats/min, a body temperature of 37.0 °C, and an oxygen saturation at room air of 94%. Her height and body weight were 154.0 cm and 53.4 kg, respectively. Fine crackles were auscultated in both lower lungs, and she had edema on both her lower extremities. She complained of bilateral hearing impairment requiring written communication, and an otolaryngology examination revealed bilateral otitis media. Notably, no findings of purpura, vision impairment, or paresthesia were detected.

Appendix A presents the laboratory data upon admission to our hospital. Urinalysis revealed severe proteinuria (3.42 g/gCr) and glomerular hematuria with excessive red blood cell cast. The blood examination revealed a further increase in serum Cre (3.59 mg/dL) and C-reactive protein (CRP) (25.2 mg/dL) levels. Concerning the vascular endothelial markers, soluble thrombomodulin and pentraxin 3 levels were high (44.1 IU/mL and 55.3 ng/mL, respectively). The serum sulfatide level was 0.78 nmol/mL and the MPO-ANCA level was 71.6 IU/mL. Additionally, proteinase-3 ANCA and anti-glomerular basement membrane (GBM) antibodies were negative. A high-resolution thoracic CT scan revealed a bilateral interstitial lung lesion and enlargement of the kidneys. Based on these findings, she was diagnosed with AAV, specifically microscopic polyangiitis, accompanying rapidly progressive glomerulonephritis (RPGN) and otitis media. Her Birmingham Vasculitis Activity Score was estimated to be 23, indicating a high disease severity. However, because the coexistence of bacterial pneumonia could not be excluded given a bilateral interstitial lung lesion, antibiotic therapy was also administered.

An intravenous injection of methylprednisolone (500 mg/day for 3 days) was first administered, followed by 30 mg oral prednisone (PSL) per day as an induction therapy. Although hemodialysis (HD) treatment was induced due to her anuria and systemic fluid retention, her CRP and MPO-ANCA levels steadily improved with glucocorticoid therapy (Figure 1).

However, on the 17th day after admission, during the induction therapy, she began to exhibit a consciousness disorder (Glasgow coma scale, E4V5M5), which fluctuated throughout the day. Her blood pressure, heart rate, and body temperature were 150/78 mmHg, 110 beats/min, and 36.9 °C, respectively. A head CT scan and magnetic resonance imaging revealed no abnormal findings. However, blood examination at that time point revealed a severe decrease in platelet counts (1.5 × 10^4^/μL), hemolytic anemia with fragmented red blood cells, and a reduction in serum haptoglobin level, suggesting the development of TMA (Appendix A). Additional examination ruled out the presence of disseminated intravascular coagulation (DIC) [14], autoimmune hemolytic anemia, and primary TMA (thrombocytopenic purpura, atypical hemolytic uremic syndrome, and Shiga toxin-producing *Escherichia coli*), and the TMA findings were thought to be secondary consequences of other diseases. Despite a significant decrease in platelet count, coagulation tests including partial thrombin time and fibrinogen were within the normal range and did not meet the criteria for DIC. The absence of a positive Direct Coombs test excluded autoimmune hemolytic anemia. Negative results for anti-cardiolipin IgG and anti-beta-2 glycoprotein 1 antibodies excluded antiphospholipid antibody syndrome. Preservation of a disintegrin-like and metalloproteinase with thrombospondin type 1 motif 13 (ADAMTS13) activity coupled with a negative inhibitor excluded thrombotic thrombocytopenic purpura. The absence of the causative bacteria in a fecal culture ruled out Shiga toxin-producing Escherichia coli-associated hemolytic uremic syndrome. Although an atypical hemolytic uremic syndrome could not be definitively ruled out, the patient had no familial history of such a syndrome. Although her ANCA level had gradually decreased with glucocorticoid therapy, we considered the possibility that AAV influenced TMA development. The dosage of PSL was maintained at 30 mg per day, and eight sessions of plasma exchange were performed. However, no improvement was observed in the TMA findings, and she continuously developed a fluctuating consciousness disorder. On the 32nd day after admission, CMV antigenemia (C10/C11) assay—routinely conducted for patients undergoing glucocorticoid therapy—was positive, and CMV-positive cell count was 296/262 per 1.5 × 10^5^ leukocytes. The CMV-IgG antibody titer on admission was positive and high, at 202 IU/mL (CMV-IgM antibody data unavailable). Therefore, it is considered that CMV infection in the present case was due to CMV reactivation. There were no symptoms or laboratory findings suggestive of enteritis, pneumonia, hepatitis, or other conditions related to CMV infection. Therefore, we suspected that TMA in the patient might have been caused by CMV infection and initiated ganciclovir infusion (1.25 mg/kg after every HD treatment). Her platelet count, hemolytic anemia, and consciousness disorder improved, CMV-positive cell count decreased, and the consciousness disorder ceased after the initiation of the treatment. Based on the clinical course, she was diagnosed with AAV with TMA induced by CMV. The fluctuating consciousness disorder was considered to be a neurological manifestation of TMA. On the 67th day following admission, approximately 5 weeks after initiating ganciclovir infusion, a kidney biopsy was performed. The biopsy samples comprised a total of 30 glomeruli, and 24 of them showed cellular or fibro-cellular crescent formation (Figure 2A,B). The walls of the renal arterioles were thickened due to subendothelial edema, and the lumens were narrow and occluded (Figure 2C). An immunofluorescence test showed no deposition of IgG or IgA (Figure 2D,E). Although IgM, C3c, and fibrinogen were focally positive, this was considered to be due to non-specific findings of glomerular sclerosis (Figure 2F–H). These pathological findings were consistent with the clinical diagnosis of AAV and TMA. In addition, immunostaining for anti-CMV-antibody was negative in the crescentic glomerulus, tubular epithelial cells, and vascular endothelial cells (Figure 2I,J).

CMV-positive cell count turned negative, and ganciclovir infusion therapy was discontinued on the 74th day after admission. As the patient’s anuria and urine findings improved sufficiently, HD was discontinued on the 76th day after admission, and her serum Cre level ultimately decreased to 1.1 mg/dL. However, her bilateral hearing impairment showed no improvement. She was discharged on the 121st day after admission. At that time, her PSL dosage was tapered down to 10 mg/day, and her MPO-ANCA level decreased to 16.6 IU/mL. No further relapse of AAV, TMA, or CMV infection was observed since her discharge. The bilateral interstitial lung lesion was found to be improved on day 52 on CT after glucocorticoid and antibiotic therapy. In addition, the patient did not exhibit any specific or worsening pulmonary symptoms during her clinical course.

### 2.2. Comparison of Three Vascular Injury Markers among the Control, Patients with AAV, and Current Case

Based on study flow 1 (Figure 3), we compared donor candidates of living kidney transplantation as a control, patients with AAV, and the current case (Table 1). Particularly, to explore the three vascular endothelial markers—serum sulfatides, soluble thrombomodulin, and pentraxin 3—as markers for TMA, we compared their levels (Figure 3). Our findings indicated that serum sulfatides in the current case were lower compared to those observed in the other two groups, while the distribution of soluble thrombomodulin and pentraxin 3 displayed notable variability (Figure 4). Regarding the components of serum sulfatides, lysosulfatides-d18:2, -d18:1, and -t18:0 levels were also lower in the present case than those of AAV patients (Table 1, Figure 4). In addition, the percentages of lysosulfatides-d18:2 and -t18:0 were lower, and lysosulfatides-d18:0 was higher in the present case than those in patients with AAV (Figure 5). Furthermore, although the serum sulfatides level was extremely low at 0.78 nmol/mL on admission, it increased to 5.66 nmol/mL at the time of kidney biopsy when the symptoms of TMA improved and inflammation due to AAV was relieved. As a result, serum sulfatides may be a potential marker of TMA; however, distinguishing a significant difference between the AAV group and the current case proved challenging for soluble thrombomodulin and pentraxin 3.

### 2.3. Association between Three Vascular Endothelial Markers and CMV-Viremia

Based on study flow 2 (Figure 6), of the 15 patients who developed CMV viremia, six patients who developed CMV infection were included (all of the patients who developed CMV infection were included in the patients who developed CMV viremia). Soluble thrombomodulin levels in patients who developed CMV viremia were significantly higher than those in patients who did not develop CMV viremia (Appendix A). Meanwhile, no significant differences were detected in serum sulfatides, their components, and pentraxin 3 levels between patients who developed CMV viremia and those who did not (Appendix A). We analyzed the association between these three vascular endothelial markers and CMV viremia; however, our analysis did not reveal a statistically significant association between CMV viremia and serum sulfatides, their components, or pentraxin 3 by univariate logistic regression analyses (Figure 7). Nonetheless, a high level of soluble thrombomodulin was significantly associated with the development of CMV viremia. We showed the result of a comparative analysis of serum sulfatides, soluble thrombomodulin, and pentraxin 3 levels among four groups: control, patients with AAV without CMV viremia, patients with AAV and CMV viremia, and the present case (Figure 8). Serum sulfatides between patients with AAV with CMV viremia and those without CMV viremia were comparable. Although the soluble thrombomodulin level in patients with AAV with CMV viremia was higher than that in patients with AAV without CMV viremia, that un patients with AAV with or without CMV viremia and that in the present case were comparable.

## 3. Discussion

We encountered a case of AAV wherein TMA developed during the induction therapy for AAV. Although TMA did not respond to plasma exchange treatment, dramatic improvement was observed with treatment for CMV infection. This implies that the TMA in the current case was CMV-induced.

Occasionally, TMA manifests in patients with severe autoimmune diseases. Manenti et al. reported that 26% of patients with AAV experienced a concurrence of TMA [5]. They also mentioned that the severity of AAV is associated with TMA development, suggesting the possibility that severe AAV directly triggers TMA development. Moreover, the patient in this present case was affected by severe AAV accompanied by RPGN and otitis media. Thus, AAV itself might have caused the development of her TMA. TMA in the present case developed 2 weeks after AAV development, when the serum ANCA titer and CRP level had decreased enough with glucocorticoid therapy. Additionally, the treatment for AAV, including eight sessions of plasma exchange, did not improve TMA in the patient. These findings suggest that the worsening of AAV was not the underlying cause of TMA in this case. Considering the clinical course of the present case, the timing of onset is a critical factor in identifying the cause of TMA in patients with AAV. Therefore, we investigated previous case reports that described detailed clinical courses and characteristics of patients who developed TMA during AAV (Table 2). Consequently, we identified 16 patients with AAV [15,16,17,18,19,20,21,22,23,24,25,26,27,28,29,30], nine of whom developed TMA during the induction phase of AAV treatment. Notably, CMV infection during the clinical course was mentioned in only one case. However, this previous report did not demonstrate a clear association between CMV infection and TMA. Among the nine patients who developed TMA during the induction phase and underwent strengthened immunosuppressive therapy and plasma exchange, four either died or developed end-stage kidney disease. In five of the nine patients who developed TMA during the induction phase, the severity of AAV had been improving at the time of TMA onset compared to the time of AAV onset. These findings suggest that some cases of TMA during induction therapy are developed by factors other than AAV itself.

In the present case, TMA was improved by the treatment for CMV. This finding implies that the TMA developed due to CMV. Additionally, CMV is known to be mostly activated several weeks after the induction into the immunosuppressed state [31,32]. Therefore, some cases of patients with AAV who developed TMA during the induction phase might be associated with CMV infection. To the best of our knowledge, there have been no case reports about patients with AAV accompanied by CMV-induced TMA. However, several studies have explored TMA occurrence among CMV-infected transplant recipients or patients with human immunodeficiency virus infection, implicating CMV infection as one of the crucial contributing factors in the development of TMA among patients with immunosuppression [33,34,35,36]. Furthermore, these studies indicated that a large part of these TMA cases were resolved with treatment for CMV, emphasizing the importance of testing for CMV infection when dealing with secondary TMA. The treatment strategy for TMA caused by AAV worsening involves strengthening immunosuppression and plasma exchange. Conversely, for TMA cases caused by CMV, immunosuppression alleviation and antiviral therapy are considered. These treatment strategies are diametrically opposed regarding the intensity of immunosuppressive therapy. Thus, while the accumulation of further evidence is required, CMV infection should be considered as a possible trigger of TMA development in patients with AAV, especially during induction therapy.

Regarding the evaluation of potential markers for CMV infection-associated TMA, we investigated three vascular endothelial injury markers—serum sulfatides, soluble thrombomodulin, and pentraxin 3—which are novel or well-known markers of vascular endothelial injury [10,11,12,13]. Our results demonstrated that only serum sulfatides may be a potential marker for CMV infection-associated TMA in patients with AAV. In addition, the patient’s serum sulfatides level was increased after TMA improvement. As for the components of serum sulfatides, lysosulphatides-d18:2, -d18:1, and -t18:0 were lower in the present case than in patients with AAV and the control. Although the percentages of these components were different between the present case and AAV, because the value of serum sulfatides was extremely low, finding the impact of these differences will be challenging. It is worth noting that serum sulfatides are surface constituents of platelets and play a pivotal role in platelet adhesion and aggregation [37,38]. This process involves the interaction of sulfatides on the platelet surface with P-selectin on neighboring platelets, ultimately facilitating the formation of large and stable platelet aggregates. Taken together, our findings suggest that serum sulfatides could potentially serve as a valuable indicator for the development of CMV infection-associated TMA in patients with AAV. However, we require additional data to validate our findings.

Regarding the mechanism of TMA development in patients with CMV infection, CMV can directly damage the endothelial cells and create microthrombi by inducing the expression of endothelial adhesion molecules and the release of von Willebrand factor [9]. Although it is still controversial whether CMV itself can really trigger TMA development, we must consider the potential for CMV-induced TMA, especially among patients with immunosuppression, such as those with severe autoimmune diseases. There is no report of CMV-induced TMA in patients with AAV, as mentioned above; however, Shiraishi et al. have reported that a case of anti-GBM disease developing TMA improved with treatment for CMV infection [39]. Their report suggested that the development of TMA might have been due to vascular endothelial injury caused by anti-GBM glomerulonephritis—a systemic vasculitis—combined with CMV-induced endothelial injury within the blood vessels. Considering that AAV is a systemic vasculitis similar to anti-GBM disease [40], the present case and previous reports imply the possibility that a combination of two distinct pathways of vascular endothelial injury may contribute to the onset of TMA. Biomarkers such as soluble thrombomodulin and pentraxin 3 are indicative of vascular endothelial damage, and their levels rise in systemic vascular disorders like DIC and vasculitis [41,42]. In the present case, soluble thrombomodulin and pentraxin 3 levels upon admission were exceptionally elevated; in addition to this, the kidney biopsy specimen also demonstrated the existence of severe vascular endothelial injury. Therefore, the present case may support the hypothesis that two different causes of endothelial injury might have potentially contributed to the onset of TMA.

As we described above, CMV can infect a variety of cells in the human body, may be latently infected in vascular endothelial cells, and may be associated with the development of severe vascular injury. In addition, severely injured vascular endothelial cells due to AAV may be at risk of CMV infection following TMA, already caused by CMV itself. Therefore, we evaluated the CMV infection in the kidney biopsy specimen and statistically analyzed the association between these three vascular endothelial markers and CMV-related outcomes such as CMV viremia. Regarding the CMV infection in the kidney biopsy specimen, as presented in Figure 2, we did not detect any CMV-infected cells. However, because the kidney biopsy was performed after TMA was improved, it is possible that CMV-infected cells were not detected. With respect to the association between the vascular endothelial markers and CMV-related outcomes, a high level of soluble thrombomodulin was significantly associated with the development of CMV viremia. Previous reports showed that soluble thrombomodulin levels were increased in patients with CMV infection and thromboembolic disease [43]. Another study demonstrated an independent association between thrombosis and acute CMV infection [44]. Additionally, chronic viral diseases such as CMV infection may lead to the transformation of endothelial cells to a procoagulant state [45]. However, the causal relationship between a high level of soluble thrombomodulin and the development of CMV viremia remains unknown. Although serum sulfatides were not significantly associated with CMV viremia, this may mean that serum sulfatides are a potential marker for not only CMV-induced TMA but also other causes of TMA in AAV patients. Nonetheless, further investigation is warranted.

Previous studies suggested predictive factors of CMV viremia during the clinical course of AAV. Low levels of total protein, hemoglobin, platelet count, lymphocytopenia, a high dose of glucocorticoid therapy, and oral candida infection may be useful markers for the prediction of CMV viremia during the treatment of AAV [8,46,47,48]. However, these studies did not extensively discuss any specific predictive markers associated with CMV-AAV or CMV-AAV-TMA. In addition, the abovementioned clinical factors suggested by these studies may indicate only immunological or nutritional status. In the present investigation, we focused on a new category, vascular endothelial injury, and its association with the occurrence of CMV and AAV or CMV, AAV, and TMA. To the best of our knowledge, this is the first attempt and further investigation of similar cases is warranted.

Consequently, serum sulfatides may potentially serve as markers for TMA during the clinical course of AAV. While soluble thrombomodulin levels were similar between AAV and the present case, a high level of soluble thrombomodulin might be linked to CMV viremia.

## 4. Materials and Methods

### 4.1. Study Participants

The dataset utilized in our previous study was analyzed here [8,13]. Candidates for living-donor kidney transplantation were analyzed as a control and 26 cases of newly developed AAV who underwent kidney biopsy without TMA were analyzed as AAV cases. The details of the data were previously described [8,13]. The current study was approved by the institutional review board of the ethical committee of Shinshu University School of Medicine (approval number: 6036) and was conducted in accordance with the principles of the Declaration of Helsinki.

### 4.2. Measurement of Serum Sulfatides

Serum sulfatides were measured from preserved sera that were obtained at the time of hospital admission of each patient using a MALDI-TOF MS system, as detailed in our prior investigation [49], with slight modifications. Initially, 50 µL of sera from individuals were mixed with 18 volumes of n-hexane/isopropanol solution (3:2, *v*/*v*) as outlined previously [49], to extract total lipids. Pooled normal human sera (#12181201, lot#BJ10633A, Cosmobio, Tokyo, Japan) were employed for the standard sulfatide quantification, and their total lipids were extracted using the same procedure as described above. Preceding this study, the sulfatide concentration in the pooled human sera was determined. Subsequently, lipid extracts from both the samples and standard sera underwent treatment with methanolic sodium hydroxide while being heated, converting sulfatides to their corresponding lysosulfatides (LS: sulfatides without fatty acids). Purification of the LS samples was achieved using Mono-tip C18 cartridges (GL Sciences, Tokyo, Japan), and equal amounts of a calibrator, N-acetylated LS-sphinganine (LS-d18:0-NAc), were added to each sample. After drying, the LS samples were dissolved in 9-aminoacridine matrix solution (5 mg/mL of 80% methanol; #92817, Merck, Darmstadt, Germany), and 1 µL of the samples were spotted on a MALDI-TOF MS plate. Analysis of LS molecules via MALDI-TOF MS was conducted using a TOF/TOF 5800 system (AB Sciex, Framingham, MA, USA) in negative-ion reflector mode, employing a 2-point external calibration with the calibrator LS-d18:0-NAc ([M–H]-584.310) and LS-(4E)-sphinganine (d18:1) ([M–H]-540.284) peaks. As in the previous study, seven LS species were detected in normal human sera, with LS-sphingadienine (d18:2), d18:1, and phytosphingosine (t18:0) identified as the major species, comprising over 80% of all LS species [49]. Finally, the sum of the concentrations of LS-d18:2, d18:1, d18:0, and t18:0 was defined as serum sulfatides.

### 4.3. Measurement of Soluble Thrombomodulin and Pentraxin 3

Soluble thrombomodulin (a marker of vascular endothelial injury) and pentraxin 3 (a marker of vascular endothelial injury and inflammation) were evaluated using the chemiluminescent enzyme immunoassay method (LSI Medience, Corporation, Tokyo, Japan) [41,50]. These markers were measured from preserved sera that were obtained upon hospital admission for each patient.

### 4.4. Analysis of the Association between CMV Viremia and Three Vascular Endothelial Injury Markers

We evaluated the association between CMV viremia and the three vascular endothelial markers, serum sulfatides, soluble thrombomodulin, and pentraxin 3, using univariate logistic regression analyses. The data analyses were based on the dataset used in the previous study [8]. From the dataset, 25 patients with AAV who were evaluated for CMV-pp65-antigen-positive cell count were analyzed. CMV viremia was defined as positivity in CMV-pp65-antigen-positive cell count. CMV infection was defined as fever, solid organ injury, and/or a hematological disorder due to CMV.

### 4.5. Statistical Analyses

The continuous variables exhibiting a normal distribution are presented as means and standard deviations, whereas those exhibiting a non-normal distribution are presented as medians and interquartile ranges. Categorical variables are presented as numbers (n) and percentages (%). Between-group data comparisons among control, patients with AAV, and the present case are presented as median and standard deviation. *p* < 0.05 was defined as significant. Analyses were performed using EZR ver. 1.60 (Saitama Medical Center, Jichi Medical University, Saitama, Japan), which is a graphical user interface for R (The R Foundation for Statistical Computing, Vienna, Austria) [51].

## 5. Conclusions

CMV infection should be considered as a potential cause of TMA in patients with AAV who develop TMA during the induction phase. Considering the mechanism of developing TMA, severe vascular endothelial injury caused by AAV and additional vascular endothelial injury resulting from factors such as CMV infection might cause TMA. In addition, serum sulfatides may be associated with TMA during the clinical course of AAV, and soluble thrombomodulin is associated with the development of CMV viremia. To validate these findings, further investigation of similar cases is warranted.

## Figures and Tables

**Figure 1 ijms-25-00812-f001:**
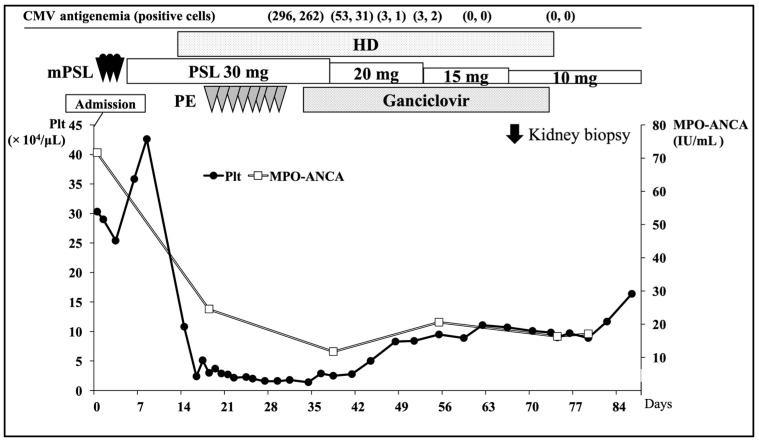
Clinical time course of the present case. CMV; cytomegalovirus, HD; hemodialysis, MPO-ANCA; myeloperoxidase anti-neutrophil cytoplasmic antibody, mPSL; methylprednisolone, PE; plasma exchange, Plt; platelet, PSL; prednisolone.

**Figure 2 ijms-25-00812-f002:**
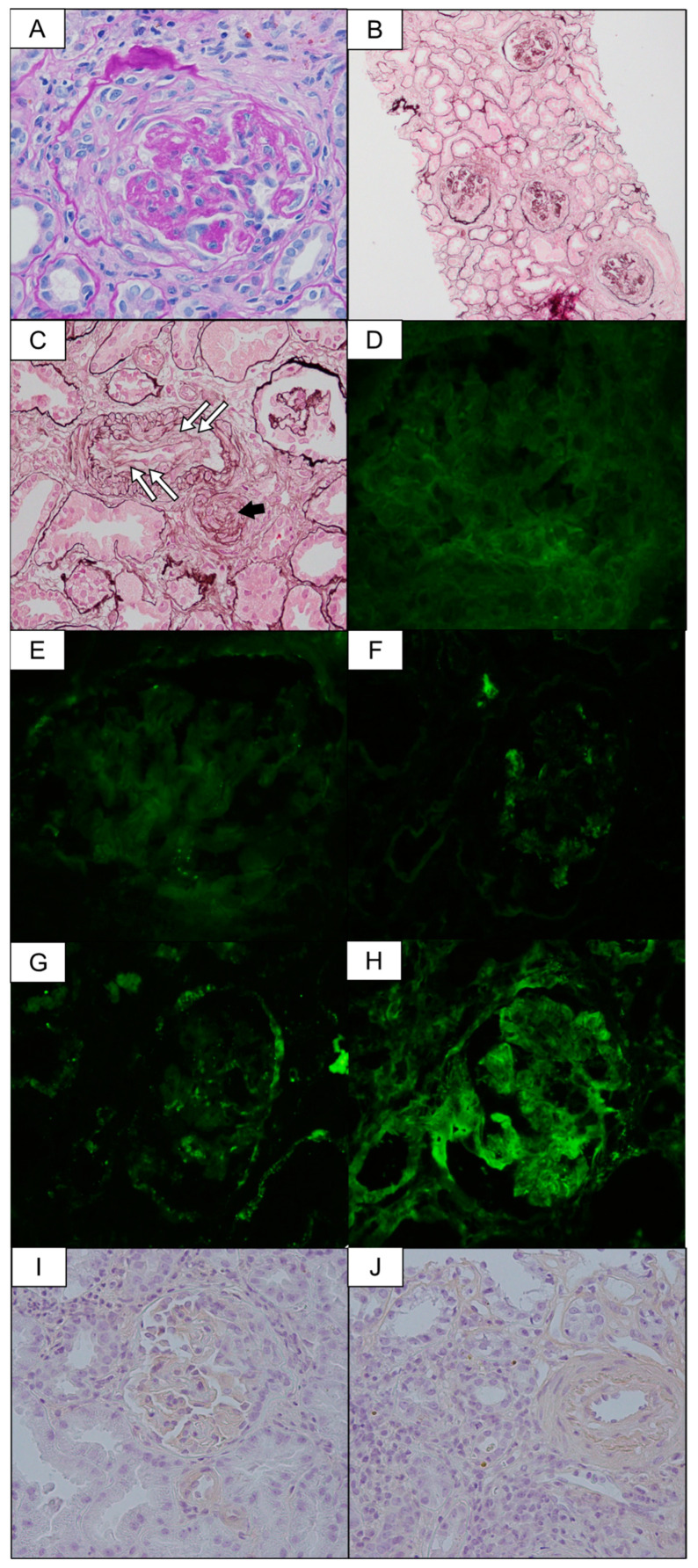
(**A**,**B**): Kidney biopsy shows a glomerulus with a fibrocellular crescent (Periodic acid Schiff stain, 40× and 10×). (**C**): The renal arterioles walls are thickened due to subendothelial edema (white arrows), and the lumens are narrow and occluded (black arrow) (periodic acid-methenamine-silver stain, 40×). (**D**,**E**): The immunofluorescence test shows no deposition of IgG or IgA (40×). (**F**–**H**): IgM, C3c, and fibrinogen are nonetheless focally positive (40×). (**I**): CMV-infected cells are not detected in the crescentic glomerulus (40×). (**J**): CMV-infected cells are not detected in the tubular epithelial cells and small arteries (40×). Positive control of anti-CMV-antibody staining is presented in Appendix A.

**Figure 3 ijms-25-00812-f003:**
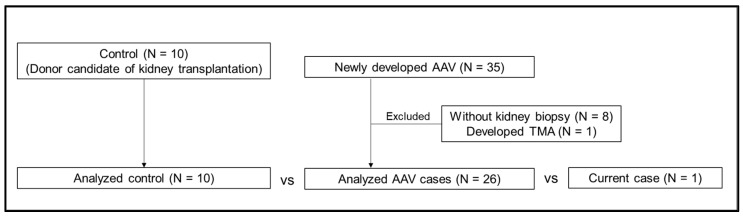
Study flow of the comparison of clinical characteristics and three vascular endothelial injury markers among three groups: control, patients with AAV, and the present case. AAV; anti-neutrophil cytoplasmic antibody-associated vasculitis, TMA; thrombotic microangiopathy.

**Figure 4 ijms-25-00812-f004:**
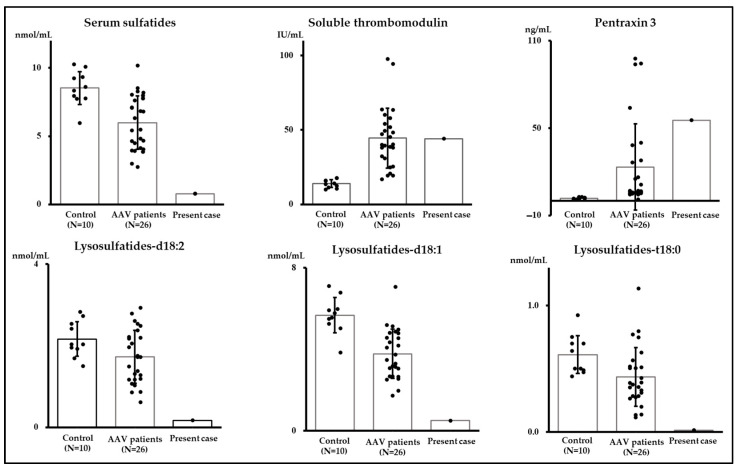
Comparative analysis of serum sulfatides, soluble thrombomodulin, and pentraxin 3 levels among three groups: control, patients with AAV, and the present case. AAV; anti-neutrophil cytoplasmic antibody-associated vasculitis.

**Figure 5 ijms-25-00812-f005:**
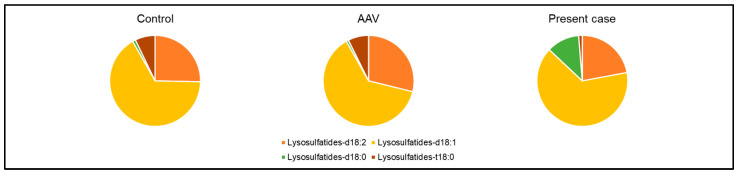
Comparative analysis of serum sulfatides’ components among three groups: control, patients with AAV, and the present case. AAV; anti-neutrophil cytoplasmic antibody-associated vasculitis.

**Figure 6 ijms-25-00812-f006:**
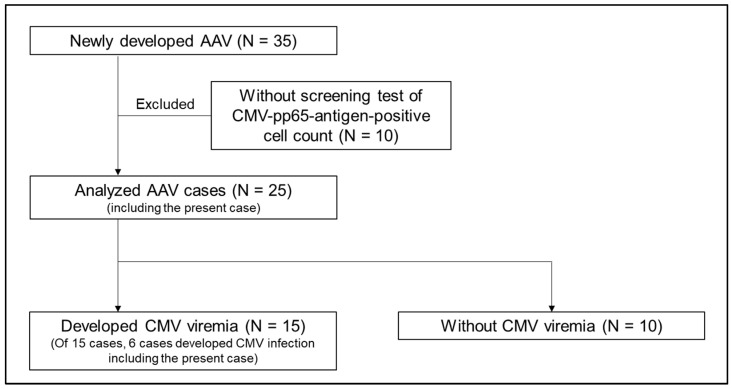
Study flow of the comparison of clinical characteristics and three vascular endothelial injury markers between patients with AAV who developed CMV viremia and those who did not. AAV; anti-neutrophil cytoplasmic antibody-associated vasculitis, CMV; cytomegalovirus.

**Figure 7 ijms-25-00812-f007:**
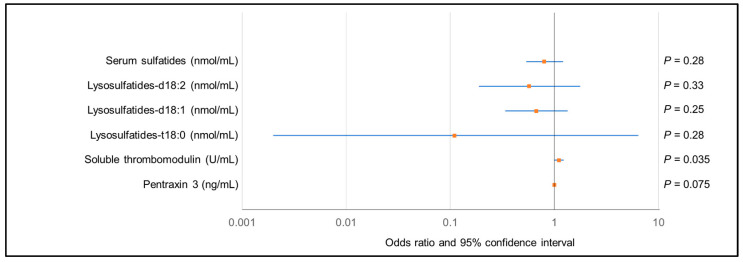
Association between CMV viremia and three vascular endothelial injury markers: serum sulfatides (and their components), soluble thrombomodulin, and pentraxin 3.

**Figure 8 ijms-25-00812-f008:**
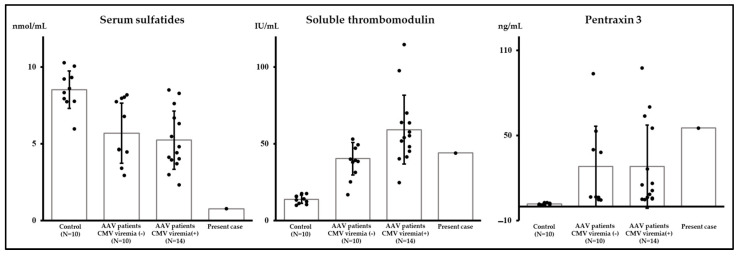
Comparative analysis of serum sulfatides, soluble thrombomodulin, and pentraxin 3 levels among four groups: control, patients with AAV without CMV viremia, patients with AAV with CMV viremia, and the present case. AAV; anti-neutrophil cytoplasmic antibody-associated vasculitis, CMV; cytomegalovirus.

**Table 1 ijms-25-00812-t001:** Comparisons of clinical characteristics and three vascular endothelial injury markers among three groups: control, patients with AAV, and the present case.

	Control (N = 10)	AAV (N = 26)	Present Case (N = 1)
Age	57 (8)	71 (10)	61
Male (n,%)	3 (30.0)	14 (53.8)	0 (0.0)
Body mass index (kg/m^2^)	23.4 (3.4)	22.6 (4.0)	22.5
Total protein (g/dL)	7.1 (0.4)	6.6 (0.7)	5.7
Albumin (g/dL)	4.3 (0.3)	2.9 (0.6)	1.1
eGFR (mL/min/1.73 m^2^)	72.3 [67.5, 76.6]	14.0 [9.8, 27.7]	10.9
C-reactive protein (mg/dL)	0.04 [0.01, 0.11]	0.91 [0.17, 5.87]	25.26
White blood cell count (/μL)	5480 [4333, 7463]	6925 [5810, 10,200]	4828
Hemoglobin (g/dL)	14.1 (1.2)	10.0 (1.5)	6.8
Platelet count (×10^4^/μL)	26.0 [23.7, 29.7]	31.1 [23.1, 37.0]	30.3
Total cholesterol (mg/dL)	222 (39)	185 (33)	104
Triglyceride (mg/dL)	153 [81, 184]	117 [97, 167]	110
Serum sulfatides (nmol/mL)	8.53 (1.28)	5.99 (2.00)	0.78
Lysosulfatides-d18:2 (nmol/mL)	2.16 (0.44)	1.73 (0.66)	0.17
Lysosulfatides-d18:1 (nmol/mL)	5.67 (0.91)	3.78 (1.22)	0.50
Lysosulfatides-d18:0 (nmol/mL)	0.09 (0.11)	0.05 (0.09)	0.09
Lysosulfatides-t18:0 (nmol/mL)	0.61 (0.16)	0.44 (0.24)	0.01
sTM (U/mL)	13.8 [11.6, 15.9]	40.2 [31.3, 53.6]	44.1
Pentraxin 3 (ng/mL)	1.6 [1.2, 2.2]	6.8 [5.3, 27.5]	55.4

Continuous variables exhibiting a normal distribution are presented as mean and standard deviation, while those exhibiting a non-normal distribution are presented as median and interquartile range. Categorical variables are presented as numbers (n) and percentages (%). AAV; anti-neutrophil cytoplasmic antibody-associated vasculitis, eGFR; estimated glomerular filtration rate, sTM; soluble thrombomodulin.

**Table 2 ijms-25-00812-t002:** Summary of reported cases of anti-neutrophil cytoplasmic antibody-associated vasculitis with thrombotic microangiopathy.

Case	Age	Sex	Diagnosis	Induction Therapy for AAV	Treatment Phase at TMA Onset	Response for Induction Therapy at TMA Onset	Treatment for TMA	Test for CMV	Prognosis
[16]	82	F	MPA	PSL	Before	-	PE, RTX	NM	Death
[18]	17	F	MPA	PSL	Before	-	PE, CY	NM	Improvement
[21]	68	F	GPA	PSL	Before		PE, RTX	NM	ESKD
[27]	32	F	MPA+SLE	PSL	Before	-	PSL	NM	Death
[28]	71	M	MPA	mPSL, PSL	Before	-	PE	NM	Improvement
[30]	70	F	MPA	mPSL, CY	Before	-	PE	NM	Death
[17]	57	M	EGPA	PSL	Induction	Good	PE, CY	NM	Improvement
[19]	66	F	GPA	mPSL, PSL, CY	Induction	Good	PE	NM	Improvement
[20]	59	F	MPA	mPSL, PSL	Induction	Good	PE, CY, RTX	NM	ESKD
[22]	77	F	MPA	mPSL, PSL	Induction	Unknown	PE, CY	NM	Improvement
[23]	59	F	MPA	mPSL, PSL	Induction	Good	PE	NM	Improvement
[24]	68	F	MPA+GBM	PE, PSL, CY	Induction	Unknown	PE, RTX	NM	ESKD
[25]	61	F	GPA	PSL	Induction	Good	PE	NM	ESKD
[26]	84	F	MPA	mPSL, PSL, RTX	Induction	Unknown	PE, ECZ	NM	Improvement
[29]	61	F	MPA+GBM	mPSL, PSL	Induction	Unknown	PE	Positive (CMV infection)	Death
[15]	51	F	MPA	-	Maintenance	-	NM	NM	Improvement

CY; cyclophosphamide, ECZ; eculizumab, EGPA; eosinophilic granulomatosis with polyangiitis, ESKD; end-stage kidney disease, F; female, GBM; glomerular basement membrane, GPA; granulomatosis with polyangiitis, M; male, MPA; microscopic polyangiitis, mPSL; methylprednisolone, NM; not mentioned, PE; plasma exchange, PSL; prednisolone, RTX; rituximab, SLE; systemic lupus erythematosus, TMA; thrombotic microangiopathy. Concerning the treatment phase at the onset of TMA, “Before” indicates before starting induction immunosuppressive treatment, “Induction” indicates during induction immunosuppressive treatment, and “Maintenance” indicates during maintenance immunosuppressive treatment. The response for induction therapy was evaluated according to the clinical course of symptoms due to anti-neutrophil cytoplasmic antibody-associated vasculitis and serum biomarkers such as ANCA titers and C-reactive protein.

## Data Availability

Data are available upon request.

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
