# Peer review of "Investigation of Clinical Features and Association between Vascular Endothelial Injury Markers and Cytomegalovirus Infection Associated with Thrombotic Microangiopathy in Patients with Anti-Neutrophil Cytoplasmic Antibody (ANCA)-Associated Vasculitis: Case-Based Research"

_ijms, 2024, doi:10.3390/ijms25020812_

Round 1

Reviewer 1 Report

Comments and Suggestions for Authors

In the article "Identification of clinical features and potential marker for
cytomegalovirus infection associated thrombotic microangiopathy in patients with anti-neutrophil cytoplasmic antibody (ANCA)-associated vasculitis: case-based research," the authors speculate that severe vascular endothelial injury caused by ANCA-associated vasculitis (AAV) and additional vascular endothelial injury resulting from factors such as cytomegalovirus (CMV) infection might contribute to thrombotic microangiopathy (TMA). Furthermore, the authors suggest that serum sulfatides may potentially serve as markers for TMA during the clinical course of AAV, and soluble thrombomodulin is a potential predictor of CMV viremia.

Comments:

1)      On page 4 (lines 137-140), the authors described that "Additional examination ruled out the presence of disseminated intravascular coagulation (DIC) [14], autoimmune hemolytic anemia, and primary TMA (thrombocytopenic purpura, atypical hemolytic uremic syndrome, and Shiga toxin-producing Escherichia coli)," but they did not specify which laboratory exams were performed to achieve this. Please address this point and make appropriate amendments to the manuscript.

2)      Provide information on CMV serostatus (IgM and IgG). Was the CMV infection a primary or secondary infection?

3)      The kidney biopsy showed crescentic glomerulonephritis (24/30 glomeruli with cellular and fibro-cellular crescents). Despite this pattern, the patient presented a remarkable improvement with only steroid treatment. According to AAV guidelines, maintenance treatment comprises cyclophosphamide, rituximab, among others. Could the authors explain this unexpected evolution? Additionally, provide more information on the tubule-interstitial and vascular compartments.

4)      What was the progression of the pulmonary symptoms and radiologic abnormalities?

5)  Why did the hearing loss not improve?

Author Response

Responses to the reviewers

We appreciate the effort and time dedicated to reviewing our manuscript. According to your comments, we have tried to respond to all your concerns to improve our manuscript as much as possible.

Reviewer 1

In the article "Identification of clinical features and potential marker for cytomegalovirus infection associated thrombotic microangiopathy in patients with anti-neutrophil cytoplasmic antibody (ANCA)-associated vasculitis: case-based research," the authors speculate that severe vascular endothelial injury caused by ANCA-associated vasculitis (AAV) and additional vascular endothelial injury resulting from factors such as cytomegalovirus (CMV) infection might contribute to thrombotic microangiopathy (TMA). Furthermore, the authors suggest that serum sulfatides may potentially serve as markers for TMA during the clinical course of AAV, and soluble thrombomodulin is a potential predictor of CMV viremia.

Comments:

  • On page 4 (lines 137-140), the authors described that "Additional examination ruled out the presence of disseminated intravascular coagulation (DIC) [14], autoimmune hemolytic anemia, and primary TMA (thrombocytopenic purpura, atypical hemolytic uremic syndrome, and Shiga toxin-producing Escherichia coli)," but they did not specify which laboratory exams were performed to achieve this. Please address this point and make appropriate amendments to the manuscript.

Thank you very much for your valuable comments. To address the related issues, we have revised the description in the Result section as follows (Lines 144 to 154):

“Despite a significant decrease in platelet count, coagulation tests including partial thrombin time and fibrinogen were within the normal range and did not meet the criteria for DIC. The absence of a positive Direct Coombs test excluded autoimmune hemolytic anemia. Negative results for anti-cardiolipin IgG and anti-beta-2 glycoprotein 1 antibodies excluded antiphospholipid antibody syndrome. Preservation of a disintegrin-like and metalloproteinase with thrombospondin type 1 motif 13 (ADAMTS13) activity, coupled with a negative inhibitor, excluded thrombotic thrombocytopenic purpura. The absence of the causative bacteria in the fecal culture ruled out Shiga toxin-producing Escherichia coli-associated hemolytic uremic syndrome. Although an atypical hemolytic uremic syndrome could not be definitively ruled out, the patient had no familial history of such a syndrome.” Finally, the findings of thrombotic microangiopathy (TMA) were considered to be secondary to other underlying diseases.

  • Provide information on CMV serostatus (IgM and IgG). Was the CMV infection a primary or secondary infection?

To address this issue, we have added the following information to the Results section (Lines 161 to 163).

“The CMV-IgG antibody titer on admission was positive and high, at 202 IU/mL (CMV-IgM antibody data unavailable). Therefore, it is considered that CMV infection in the present case was due to CMV re-activation.” (This means it was a secondary infection).

  • The kidney biopsy showed crescentic glomerulonephritis (24/30 glomeruli with cellular and fibro-cellular crescents). Despite this pattern, the patient presented a remarkable improvement with only steroid treatment. According to AAV guidelines, maintenance treatment comprises cyclophosphamide, rituximab, among others. Could the authors explain this unexpected evolution? Additionally, provide more information on the tubule-interstitial and vascular compartments.

Regarding the treatment strategy for ANCA-associated vasculitis, as the Reviewer mentioned, glucocorticoids and immunosuppressants combination is recommended. However, elderly patients with AAV are recommended glucocorticoid monotherapy. Our patient was treated with methylprednisolone, for 3 days, and 30 mg of daily oral prednisolone therapy, subsequently. Considering prior guidelines for AAV, this treatment was not so weak. The dose of glucocorticoid therapy was appropriate to improve kidney function in the present case.

In addition, regarding the kidney biopsy specimen, 24/30 glomeruli showed cellular and fibro-cellular crescent formation. The blood vessels have been collected up to the level of interlobular arteries, and interlobular arteries with disruption or loss of elastic lamina were scattered. Additionally, lymphocytic infiltration into the vessel walls was observed in some areas, indicating small vessel vasculitis. However, there were only 2/30 glomeruli with global sclerosis. The tubule interstitial area was not severely injured. Inflammatory cells infiltrated only 10% of the tubule interstitial area and a slight fibrotic change was also observed. Thus, the fact that irreversible pathological changes had not occurred may be associated with a good kidney outcome.

  • What was the progression of the pulmonary symptoms and radiologic abnormalities?

Following your comment, we have added the following comments to the Results section (Lines 120 to 121 and 200 to 202, respectively).

“Concerning the pulmonary symptoms and radiologic abnormalities, because the coexistence of bacterial pneumonia could not be excluded given a bilateral interstitial lung lesion, antibiotic therapy was also administered.”

“The bilateral interstitial lung lesion was found to be improved on day 52 on CT after glucocorticoid and antibiotic therapy. In addition, the patient did not exhibit any specific or worsening pulmonary symptoms during her clinical course.”

  • Why did the hearing loss not improve?

According to previous studies, the functional prognosis of the ear in patients with otitis media with ANCA-associated vasculitis (OMAAV) is not favorable, particularly in terms of hearing ability (Kobari. Y. et al. Intern Med. 2017;56:3365-3367 [PMID: 29021451]). Yoshida (Otol Jpn 25: 202-207, 2015 [Japanese]) reported that the treatment outcomes of 284 cases of otitis media with ANCA-associated vasculitis were as follows: 30% of them completely recovered, 30% of them partially improved, and the remaining 40% did not improve or experienced hearing function deterioration. In addition, Yoshida also reported that, particularly in cases of OMAAV with hearing impairment, there was no significant improvement in hearing function. Another study found that younger age, male sex, shorter period from onset to diagnosis, use of IVCY, and better hearing threshold at diagnosis were associated with a better hearing prognosis in patients with OMAAV (Iwata S. et al. Auris Nasus Larynx.2021 48:377-382 [PMID: 32951931]].

Thus, the lack of improvement in hearing function in our patient is consistent with the course reported in the literature. Although kidney function recovered in the present case, unfortunately, hearing function did not. This may be due to the severe inflammatory lesion in her ears manifested during her clinical course.

Reviewer 2 Report

Comments and Suggestions for Authors

Comments:

 1)      Line 160-161; ‘Immunofluorescence test showed no deposition of antibodies or immune complexes (not shown)’, fortify the figure 2 with immunofluorescence data and also describe the same in methodology.

2)      It is highly recommended to show the 10X image of the intact kidney biopsy tissue section in figure 2, to support the statement line#157-158.

3)      Present study is the extension of previous study reference #8 & #13,where authors used same patients and evaluated Serum Sulfatide  levels. Is there duplication of the same data?

4)      Methodology is poorly explained. Give reference catalogue of kits or chemicals used.

5)      Cite appropriate references showing the association of CMV-AAV or CMV-AAV-TMA and predictive markers (if available) and discuss the same.

 Major comments:

1)      Single case based study (n=1) will not support the title “potential marker for cytomegalovirus infection associated thrombotic microangiopathy in patients with anti-neutrophil cytoplasmic antibody (ANCA)-4 associated vasculitis”, justification required in this regard. Data is not supporting the title. It is highly recommended to provide the comparison in at least n =10 CMV infected AAV patients with TMA for any conclusive remarks.

Comments on the Quality of English Language

Minor editing/reframing of sentence may require. 

Author Response

Responses to the reviewers

We appreciate the effort and time dedicated to reviewing our manuscript. According to your comments, we have tried to respond to all your concerns to improve our manuscript as much as possible.

Reviewer 2

Comments:

  • Line 160-161; ‘Immunofluorescence test showed no deposition of antibodies or immune complexes (not shown)’, fortify the figure 2 with immunofluorescence data and also describe the same in methodology.

Thank you very much for your suggestion. We have added the immunofluorescence staining images to the revised Figure 2. In addition, we have added corresponding explanations to the figure legend and relevant descriptions in the Results section as reported below (Lines 173 to 181 and 184 to 191).

“The biopsy samples comprised a total of 30 glomeruli, and 24 of them showed cellular or fibro-cellular crescent formation (Figure 2A and B). The walls of the renal arterioles were thickened due to subendothelial edema, and the lumens were narrow and occluded (Figure 2C). Immunofluorescence test showed no deposition of IgG or IgA (Figure 2D and E). Although IgM, C3c, and fibrinogen were focally positive, this was considered to be due to non-specific findings of glomerular sclerosis (Figure 2F, G, and H). These pathological findings were consistent with the clinical diagnosis of AAV and TMA. In addition, immunostaining for anti-CMV-antibody was negative in the crescentic glomerulus, tubular epithelial cells, and vascular endothelial cells (Figure 2I, 2J).”

  • It is highly recommended to show the 10X image of the intact kidney biopsy tissue section in figure 2, to support the statement line#157-158.

Following your recommendation, we have added a new image to Fig. 2 (Figure 2B) and revised the text in the Results section regarding the kidney biopsy specimen and in the related figure legend similar to the previous query (Lines 173 to 181 and 184 to 191).

  • Present study is the extension of previous study reference #8 & #13,where authors used same patients and evaluated Serum Sulfatide levels. Is there duplication of the same data?

In this study, we employed the identical reference database as cited in references 8 and 13 within the manuscript. We appropriately cited these previous studies. Furthermore, the primary analysis conducted in this study—involving comparisons of serum sulfatides, soluble thrombomodulin, and pentraxin 3 among control individuals, patients with ANCA-associated vasculitis (AAV), and the present case—represents a novel contribution. Additionally, the analysis for the association between three markers of vascular endothelial injury and cytomegalovirus (CMV) viremia is also entirely new. Therefore, we believe the results of our study are novel and never overlap with those of the previous manuscript.

4) Methodology is poorly explained. Give reference catalogue of kits or chemicals used.

Following your comment, we have revised the method used for the quantification of serum sulfatides as follows (Lines 421 to 443):

“Serum sulfatides were measured from preserved sera that were obtained at the time of hospital admission of each patient using a MALDI-TOF MS system, as detailed in our prior investigation [49], with slight modifications. Initially, 50 µL of sera from individuals were mixed with 18 volumes of n-hexane/isopropanol solution (3:2, v/v) as outlined previously [49], to extract total lipids. Pooled normal human sera (#12181201, lot#BJ10633A, Cosmobio, Tokyo, Japan) were employed for the standard sulfatide quantification, and their total lipids were extracted using the same procedure as described above. Preceding this study, the sulfatide concentration in the pooled human sera was determined. Subsequently, lipid extracts from both the samples and standard sera underwent treatment with methanolic sodium hydroxide while being heated, converting sulfatides to their corresponding lysosulfatides (LS: sulfatides without fatty acids). Purification of the LS samples was achieved using Mono-tip C18 cartridges (GL Sciences, Tokyo, Japan), and equal amounts of a calibrator, N-acetylated LS-sphinganine (LS-d18:0-NAc), were added to each sample. After drying, the LS samples were dissolved in 9-aminoacridine matrix solution (5 mg/mL of 80% methanol; #92817, Merck, Darmstadt, Germany), and 1 µL of the samples were spotted on a MALDI-TOF MS plate. Analysis of LS molecules via MALDI-TOF MS was conducted using a TOF/TOF 5800 system (AB Sciex, Framingham, MA, USA) in negative ion reflector mode, employing a 2-point external calibration with the calibrator LS-d18:0-NAc ([M–H]- 584.310) and LS-(4E)-sphinganine (d18:1) ([M–H]- 540.284) peaks. As in the previous study, seven LS species were detected in normal human sera, with LS-sphingadienine (d18:2), d18:1, and phytosphingosine (t18:0) identified as the major species, comprising over 80% of all LS species [49]. Finally, the sum of the concentration of LS-d18:2, d18:1, d18:0, and t18:0 was defined as serum sulfatides.”

5) Cite appropriate references showing the association of CMV-AAV or CMV-AAV-TMA and predictive markers (if available) and discuss the same.

Following your comment, we have added these comments to the discussion section (Lines 396 to 405).

“Previous studies suggested predictive factors of CMV viremia during the clinical course of AAV. Low levels of total protein, hemoglobin, platelet count, lymphocytopenia, a high dose of glucocorticoid therapy, and oral candida infection may be useful markers for the prediction of CMV viremia during the treatment of AAV [8,46-48]. However, these studies did not extensively discuss any specific predictive markers associated with CMV-AAV or CMV-AAV-TMA. In addition, the abovementioned clinical factors suggested by these studies may indicate only immunological or nutritional status. In the present investigation, we focused on a new category, vascular endothelial injury, and its association with the occurrence of CMV and AAV or CMV, AAV, and TMA. To the best of our knowledge, this is the first attempt, and further investigation of similar cases is warranted.”

Major comments:

  • Single case based study (n=1) will not support the title “potential marker for cytomegalovirus infection associated thrombotic microangiopathy in patients with anti-neutrophil cytoplasmic antibody (ANCA)-4 associated vasculitis”, justification required in this regard. Data is not supporting the title. It is highly recommended to provide the comparison in at least n =10 CMV infected AAV patients with TMA for any conclusive remarks.

We agree with the Reviewer’s comment. Ideally, we would like to collect relevant data on as many patients with CMV and AAV as possible. However, considering the prevalence of patients with CMV infection, AAV, and TMA, collecting a sufficient number of cases should be impossible.

Therefore, firstly, we have changed the title from “Identification of clinical features and potential marker for cytomegalovirus infection associated thrombotic microangiopathy in patients with anti-neutrophil cytoplasmic antibody (ANCA)-associated vasculitis: case-based research” to “Investigation of clinical features and association between vascular endothelial injury markers and cytomegalovirus infection associated with thrombotic microangiopathy in patients with anti-neutrophil cytoplasmic antibody (ANCA)-associated vasculitis: Case-based research”. Accordingly, we have also changed the description of the purpose and conclusion of this study in the Abstract, Introduction, and Conclusion sections (Lines 1 to 6, 34 to 36, 45 to 47, 78 to 80, and 474 to 475).

Secondly, we have added the changes in serum sulfatides following the clinical course to the Results and Discussion section.

“Although the serum sulfatides level was extremely low at 0.78 nmol/mL on admission, it increased to 5.66 nmol/mL at the time of kidney biopsy when the symptoms of TMA improved and inflammation due to AAV was relieved” (Lines 216 to 219).

“In addition, the patient’s serum sulfatides level was increased after TMA improvement” (Lines 341 to 343).

However, unfortunately, we do not have multi-time point data of soluble thrombomodulin and pentraxin 3.

Thirdly, we have prepared a new figure (Figure 8) showing a comparison of the three vascular endothelial injury markers among the control, AAV without CMV viremia, AAV with CMV viremia, and the present case. This figure may be a complement to Figures 4 and 7. We also added the explanation of Figure 8 to the Results section (Lines 255 to 262).

We thank the Editor and the Reviewers for their time and helpful feedback and hope that the changes we have made to the manuscript have brought it up to the standards for publication in the International Journal of Molecular Sciences.

Round 2

Reviewer 2 Report

Comments and Suggestions for Authors

No comments

Comments on the Quality of English Language

Minor English editing may require. Section Editor may check for same.